# Investigating the Potential of Thin Silicon Nitride Membranes in Fiber-Based Photoacoustic Sensing

**DOI:** 10.3390/s23031207

**Published:** 2023-01-20

**Authors:** Yorick Konijn, Edcel Salumbides, B. Imran Akca

**Affiliations:** LaserLab, Department of Physics and Astronomy, VU University, De Boelelaan 1081, 1081 HV Amsterdam, The Netherlands

**Keywords:** photoacoustic sensing, fiber optics, silicon nitride membrane

## Abstract

The detection of methane, a strong greenhouse gas, has increased in importance due to rising emissions, which partly originate from unreported and undetected leaks in oil and gas fields. The gas emitted by these leaks could be detected using an optical fiber-based photoacoustic sensor called PAS-WRAP. Here, we investigate the potential of silicon-based membranes as more sensitive microphones in the PAS-WRAP concept. Toward this goal, we built a setup with which the frequency response of the membranes was interrogated by an optical fiber. Multiple mounting mechanisms were tested by adapting commercial interferometry systems (OP1550, ZonaSens, Optics11 B.V.) to our case. Finally, methane detection was attempted using a silicon nitride membrane as a sensor. Our findings show a quality factor of 2.4 at 46 kHz and 33.6 at 168 kHz for a thin silicon nitride membrane. This membrane had a frequency response with a signal-to-background ratio of 1 ± 0.7 at 44 kHz when tested in a vacuum chamber with 4% methane at 0.94 bar. The signal-to-background ratio was not significant for methane detection; however, we believe that the methods and experimental procedures that we used in this work can provide a useful reference for future research into gas trace detection with optical fiber-based photoacoustic spectroscopy.

## 1. Introduction

Climate change is a grand challenge that has ended up as one of the biggest environmental hazards confronting the world today. Methane is a potent contributor to climate change, which has been trapping 84 times as much heat as carbon dioxide over 20 years [1]. Among many other sources, the oil and gas industry is the largest industrial source of methane emissions [2,3]. Knowing that they will be a part of the energy system for decades to come, it is, therefore, vital to reduce the immediate environmental impacts associated with producing and consuming these fuels. A good methane detector should be sensitive, low-cost, multiplexable, require less maintenance, and be able to operate on batteries. It should also have a minimum detection limit of 5 ppm of CH_4_ of atmospheric pressure. Among existing solutions [4,5,6,7,8,9,10,11,12,13,14,15,16], optical fiber-based solutions hold great promise for meeting the abovementioned requirements [12,13]. Recently, an optical fiber-based sensor called PAS-WRAP was developed, which uses the photoacoustic (PA) effect for gas trace detection [16]. This concept relies on optical fibers being wrapped around a chamber, working like a guitar upon acoustic excitation. The PA effect is the process of sound generation in a material resulting from the absorption of photons. Photoacoustic spectroscopy (PAS) is a well-established technique for gas trace detection [17], which has a detection sensitivity for the concentration of trace gasses of up to parts per billion, and it has a high specificity enabling the detection of individual gas species [9,10]. To investigate specific molecules, the wavelength of a narrow-band light source, such as a laser, is matched to the excitation wavelength of the molecule. Detection of the modulated sound waves generated by these molecules forms the basis of PAS. Sigrist et al. have significantly contributed to the field of PAS-based gas detection [11,12,13,14]. They have introduced the concept of pulsed resonance PAS and demonstrated the detection of NO_2_ by using four miniature commercial acoustic microphones [12]. Another new concept that was developed by the same group is called differential mode excitation photoacoustic spectroscopy, which relies on the selective excitation of two different resonant modes in a specially designed photoacoustic cell [14]. Using an electret microphone, they obtained a limit of detection of 25 ppm m^−1^ for acetone in room air. Typical photoacoustic sensors include condenser microphones [18] and quartz tuning forks [19] that rely on electronic or piezo-electric detection. Cantilever-based microphones [20,21] have also been used that rely on an optical-based detection modality, to which the PAS-WRAP fiber sensor and the membrane sensor presented here belong. The overall system design of the PAS-WRAP concept offers several advantages, such as easy multiplexing, low-maintenance, versatile detection, and low cost, which makes it a promising PA sensor for trace gas detection. However, preliminary results of the PAS-WRAP sensor demonstrated the detection of 0.2% CH_4_ in N_2_. Even though this is a very promising and novel idea, the sensitivity is not sufficient for methane detection, and it has to be increased by a factor of ~500 to reach the desired detection limit. One of the factors influencing the detection limit of a PAS-WRAP sensor is the sensitivity of the microphone. This sensitivity is often expressed as the quality factor, Q, which is the resonance frequency of the microphone divided by the full width at half maximum. In the PAS-WRAP design, an optical fiber wrapped around a chamber is used as a microphone with a Q ≈ 10. A different microphone based on a silicon nitride membrane has become a sensor of interest, as Zwickl et al. showed, a Q of >10^6^ at room temperature for a silicon nitride membrane [22]. This Q is more than the required 500× increase of the quality factor in the wrapped optical fiber and might enable the detection of methane at the desired detection limit of 5 ppm. Here, we investigate the viability of a silicon nitride membrane as an optomechanical microphone to be used in the PAS-WRAP concept. We start by characterizing the frequency response to identify the resonance frequencies of these membranes. Using the resonance frequency, we compute the quality factor. Next, we test the detection limit of the membranes in a methane detection experiment. We concluded that the fixation of the silicon membranes could dramatically change the amplitude and the frequency of the membrane resonances by changing the boundary conditions. In that respect, their application in PA sensing is limited.

## 2. Materials and Methods

### 2.1. Membranes and Membrane Holders

In this work, two types of silicon nitride membranes were used (Figure 1). The first type was gold-coated membranes produced by Silison, with a frame size of 17.5 mm × 17.5 mm and a frame thickness of 200 µm (Figure 1a). The dimensions of the Silison membranes were 10.0 mm × 10.0 mm × 100 nm. The second type of membrane was produced by Norcada, with a frame size of 10.0 mm × 10.0 mm and a frame thickness of 500 µm (Figure 1b). The dimensions of the Norcada membranes were 5.0 mm × 5.0 mm × 200 nm. They are not gold-coated as Silison membranes. The Silison membranes were chosen for this study as they were readily available in our lab. The Norcada membranes were chosen at a later stage for their reported high Q factor [19]. The membranes were stored in air-tight boxes to keep them dust-free. For ease of referral, the membranes are called Silison A and Silicon B, Norcada A and Norcada D. Two membranes with similar properties from the same vendor (i.e., Silison A and Silison B, and Norcada A and Norcada D) were used in the experiments to study the membrane-dependent variations. 

The resonance frequencies of a square membrane can be calculated using the equation [23]: (1)$$f_{n_{x},n_{y}}=\frac{1}{2L}\sqrt{\frac{S}{\rho }(n_{x}{}^{2}+n_{y}{}^{2})}$$
where *S* is the stress of the membrane and *ρ* is the density of the membrane; *n_x_* and *n_y_* are the mode shape identifiers. No datasheet was available for the Silison membranes; therefore, it is hard to predict the resonance frequency of this membrane. From the Norcada membrane datasheet, we found that *S* = 1 GPa, *ρ* = 3100 g/m^3^, and *L* = 5.0 mm. Therefore, the resonance frequency of the first mode was calculated as *f*_1,1_ ≈ 80 kHz.

Three different types of 3D-printed holders were used to keep the silicon nitride membranes in position. The first type of holder kept the membrane in place by exerting a force on the frame of the membrane with a top plate and a rubber o-ring in between, as shown in Figure 3a. In the second type, the frame of the membrane was glued to a metal washer plate to keep the membrane in place, as shown in Figure 4b.

### 2.2. Measurement Setup

The setup depicted in Figure 2 was used to measure the deflection of the membranes for sound waves at different frequencies and amplitudes. The sound waves were generated using a low-voltage piezo element (Thorlabs, PA1CE), which was glued to an aluminum block. The aluminum block was placed on a vertical translation stage (Thorlabs, VAP10/m) to control the distance between the membrane and the piezo element. The deflection of the membrane was detected by a single-mode optical fiber glued to a 3D-printed probe. The probe was attached to a micromanipulator (Scientifica, PatchStar), which was manually positioned using a control cube. The optical fiber was connected to an interferometer (Optics11, OP1550) with a tunable laser source with a central wavelength around λ = 1550 nm. The interferometer generated an output signal at a rate of 20 ksamples/s, which was sent to a DAQ (National Instruments C-2120). The DAQ converted this signal to a digital signal that was read by a self-written LabView program. Additionally, the LabView program generated a signal, which the DAQ converted to an electronic signal, to drive the piezo element. In this configuration, the duration, frequency, and amplitude of the piezo transducer were controlled by the sine waves generated by the DAQ. The lock-in amplifier is a built-in component of the OP1550 system, which was developed by Optics11. By scaling the lock-in signal and interferometric readout, a linear relation is obtained to the membrane deflection. The phase can be obtained by computing the angle between the scaled interferometric readout voltage and scaled lock-in voltage. 

To measure resonances at higher frequencies, we changed the setup and used a different interferometer (Optics11, ZonaSens). The ZonaSens uses two optical fibers for its interferometry. The first optical fiber was used as a reference arm and was connected to two fiber Bragg gratings (FBG) with wavelengths of 1550 nm and 1560 nm. The second fiber was connected to an FBG with its reflection centered at 1550 nm, which also collects part of the scattered illumination on the membrane and becomes a part of the detection arm. Additionally, a function generator (UNI-T, UTG962E) was used to drive the piezo element instead of the DAQ.

## 3. Results and Discussions

### 3.1. Measurements with O-Ring Membrane Mount

Figure 3 shows the spectra obtained from Silison A membrane using the o-ring clamping method. Bolts were used to put pressure on the frame of the membrane. In the experiment, the marked bolts, shown in Figure 3a, were tightened by four full rotations in a set order: (1) lower left, (2) upper right, (3) upper left, and (4) lower right. From this position, the bolts could be tightened by a quarter turn (+90°) or unwound (−90° and −180°). To check the repeatability of the spectrum at a specific bolt rotation, three consecutive spectra were recorded at 1 min intervals for a positive quarter turn of the bolts (Figure 3b). The spectra of the Silison membrane displayed in Figure 3b have multiple peaks in the region of 1.8–3.6 kHz, which corresponds to a region where the audible environmental noise is dominant. The frequency response in this area contains more variance compared to the peaks above 4 kHz. As can be seen in Figure 3c, rotating the bolts resulted in changes in the frequency response of the membrane. For example, unwinding the bolts by 180° resulted in two frequency peaks at 4 kHz and 6.5 kHz, which do not show up for the other bolt rotations. Figure 3d shows a repetition of the experiment on a different day, for which the membrane was remounted onto the holder. Slight differences in the spectrum are visible; in particular, a peak around 7.5 kHz is visible for the repeated experiment at −90°. We took each spectrum with the minimum possible frequency step (~10 Hz) to be able to resolve peaks with higher details.

**Figure 3 sensors-23-01207-f003:**
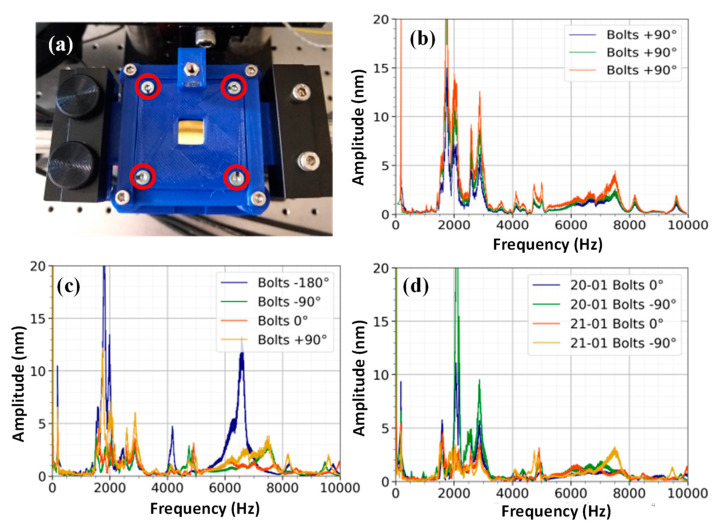
(**a**) The sandwiched membrane in between two plates with an o-ring and four bolts. The pressure exerted by the o-ring depends on how much the bolts, circled in red, are tightened. (**b**) Three consecutive measurements of the membrane at a specific bolt rotation. Measurements were taken at 1 min intervals, and they show a consistent frequency response. (**c**) The frequency response of the membrane is mapped at the center of the membrane for varying bolt rotations. The baseline rotation of 0° is set to four full rotations. (**d**) Spectra of the membrane are compared for a repeat of the experiment on the next day. The piezo element was used to generate a frequency chirp pattern with a duration of 150 s at an amplitude of 10 Vpp. The distance from the piezo to the membrane was set at approximately 5 mm.

### 3.2. Measurements with Washer Ring Membrane Mount

Figure 4a,b show gold-coated Silison A membrane, which is mounted by gluing the frame to a washer plate. Repeated spectral measurements display a peak at 9 kHz for this specific membrane, as shown in Figure 4c, where three consecutive measurements of the same membrane under the same boundary conditions are given. The quality factor of this resonance peak was obtained by locating the maximum amplitude within a frequency range of 8–10 kHz. Next, the full width at half maximum was computed by selecting the first and last component smaller than half the peak value. An average quality factor of 28 ± 1.2 was obtained from six spectral measurements. Figure 4d displays measurements of the same membrane that were repeated over multiple days. The measurement on the second of March (2-March) and the third of March (3-March) indicate minor deviations in frequency response. For example, the 2-March measurement displays two peaks at 1.3 kHz and a peak at 3.96 kHz, which are not present in the 3-March measurement. An additional experiment was performed on the 18th of March (18-March), in which the placement of the piezo on the set-up was changed compared to the previous measurements (Figure 4d). In previous measurements, the response of the piezo was non-flat due to being glued to a plastic tube. During the 18-March measurement, the piezo was glued to an aluminum block, which resulted in a flat frequency response between 0–10 kHz. The piezo glued to the aluminum block was used for the experiments from this section onward. Because of the difference in the mounting of the piezo, differences in the spectrum of the 18-March measurement in Figure 4d were not attributable to inconsistencies in the mounting mechanism.

**Figure 4 sensors-23-01207-f004:**
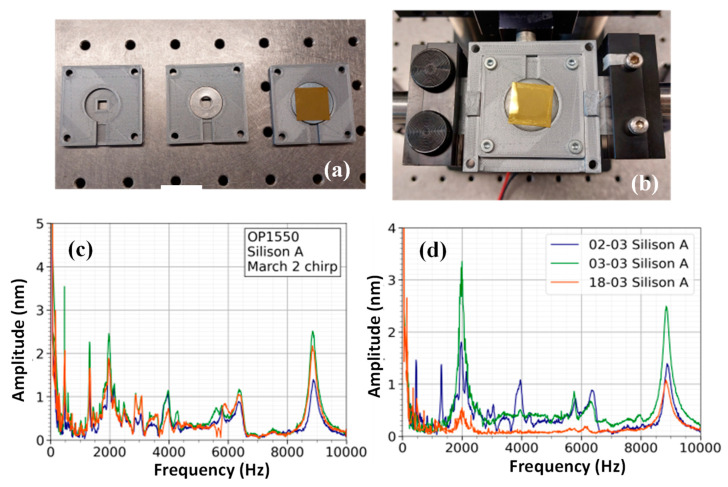
(**a**,**b**) The Silison A membrane glued to a washer ring. The spectra show a consistent peak at 8850 ± 10 Hz for experiments on the same day (**c**) and between days (**d**).

### 3.3. Membrane Measurements with OP1550 Interferometer

The spectra of the available membranes are shown in Figure 5. The Silison A membrane has a clear resonance peak at 9 kHz; however, the spectra of the Silison B, Norcada A, and Norcada D membranes do not display a clear resonance peak in the frequency range of 0–10 kHz. As a result of the limited output rate (20 ksamples/s), no frequencies beyond 10 kHz can be detected with the OP1550 interferometer. The estimated resonance frequency for the Norcada membranes was 80 kHz, and therefore, the OP1550 interferometer cannot be used to measure this resonance frequency. The Norcada membranes were quite fragile; therefore, they were not mounted on a washer plate because of the risk of breaking the membrane when gluing the frame to the washer plate.

### 3.4. Membrane Measurements with ZonaSens

The ZonaSens was used to investigate the frequency response of the silicon nitride membranes up to 200 kHz. This interferometer is automatically calibrated using internal software (SmartSonic). Spectra of three of the silicon nitride membranes were measured with ZonaSens. These spectra are shown in Figure 6. The Silison membrane was measured twice to investigate the reproducibility of the frequency response. Interestingly, the spectra of all three membranes have a frequency peak at 168 kHz, which indicates that part of the setup might induce a response at this frequency. The Silison A membrane has a small peak at 9 kHz, which is consistent with the previous results. In the spectrum of this membrane, two additional peaks containing multiple data points are visible in Figure 6a. The first peak is at a frequency of 66 kHz. In the first Silison A spectrum (blue), this peak has FWHM = 11 kHz and a Q = 6. The second spectrum (green) has FWHM = 9 kHz and Q = 7. The second peak is at a frequency of 140 kHz with FWHM = 4 kHz and Q = 35 (blue), and FWHM = 6 kHz and Q = 24 (green). In Figure 6b, the spectra of two Norcada membranes are shown. Most of the peaks of these spectra occur at the same frequency. The peak at 46 kHz appears in both spectra. For the Norcada A membrane, this peak has FWHM = 5 kHz and Q = 9.2. For the Norcada D membrane, this peak has FWHM = 8 kHz and Q = 5.8. Other peaks are visible in the spectra of the Norcada membranes; however, most of these peaks consist of a single data point. To increase the frequency resolution of the spectra, a linear frequency chirp excitation pattern can be implemented.

### 3.5. Methane Detection

A methane detection experiment was attempted using the setup given in Figure 7a. In these experiments, the Norcada D membrane was used. A custom-made gas chamber was used, and the methane concentration was set at 4% at a pressure of 0.94 bar. An excitation laser that was matched to a rovibrational line of methane at 1650.9 nm, was guided toward the membrane via an optical fiber. The laser source (Eblana Photonics) was modulated to tune the sound waves produced by the photoacoustic effect. The frequency response of the Norcada D membrane is given in Figure 7b. However, these peaks coincide with increased levels of background noise. For example, the 44 kHz peak has an amplitude of 0.03 ± 0.017 nm and has a signal-to-background ratio (SNR) of 1 ± 0.7. Similarly, the 116 kHz peak has an amplitude of 0.04 ± 0.012 nm with an SNR of 1 ± 0.3. No signal with a significant difference from the background was observed. The SNR is calculated by dividing the amplitude of the signal at the resonance frequency by the amplitude of the background that is relatively far from the resonance peak. 

## 4. Discussions

The findings from this study suggest that a silicon nitride membrane might not be a viable microphone for methane detection with a fiber-based photoacoustic spectrometer. The viability was investigated by characterizing the frequency responses of the Silison and Norcada membranes using the OP1550 and ZonaSens interferometers. These interferometers were able to characterize the spectrum of these membranes up to 10 kHz (OP1550) and 200 kHz (ZonaSens). The frequency response was used to locate the resonance frequency, which we hoped would be consistent and have a high-quality factor. However, the measured resonance frequencies were generally inconsistent, as the observed spectra were not fully reproducible for repeated experiments with the same membrane and mounting mechanism. We have observed several peaks for each membrane, which represent the eigenfrequencies of the membranes. Some of them are weak, but they can still be excited, as we observed. Experiments in which the detection of 4% methane was attempted with a Norcada membrane showed no significantly different signals from the background signal.

Inconsistency in the frequency responses of the membranes could be due to several reasons. First, the surface tension of the membranes could change between consecutive measurements. Fletcher et al. showed that the tension is influenced by the temperature of the membrane [24]. Thus, a difference in room temperature could explain the inconsistency in the observed frequency responses. As a solution, the resonance spectrum of the membranes can be measured in a closed gas chamber to avoid the influence of room temperature variation. However, we could not perform these measurements since our current gas chamber is too small to place the measurement setup.

Secondly, the changes in frequency response were induced by changes in the pressure exerted on the frame of the membrane in the o-ring clamping mechanism. In this mechanism, the pressure exerted on the frame might induce a change in the tension of the membrane. However, our results on this tension shift remain inconclusive, as experiments on this effect were limited to a single membrane that displayed no resonant frequency in the spectrum. Knowing the distribution of stress over the membranes for different mounting configurations can give some insight into their effect. The best approach is to measure it experimentally; however, it can be difficult. Alternatively, one can simulate the stress distribution using finite element modeling simulations. However, in our case, the relatively big size of the membranes and their frames made it quite difficult to simulate the actual devices. Moreover, the exact mechanical properties of some of the components (frame, bolts, rubber spacing, etc.) are not known, which can affect the accuracy of the results significantly. As common sense, when a single bolt is tightened, we expect that the maximum of the membrane stress shift toward the edge of the membrane where the corresponding bolt is closer. When all bolts are tightened in the same way, then the membrane stress is uniformly distributed over the membrane. In our experiments, we tried to create a uniform stress distribution over the membrane to be as repeatable as possible. However, we concluded that it is quite a challenging task. 

Another source of inconsistency in the frequency response might be due to changes in the alignment of the setup between consecutive measurements. The piezo transducer was roughly aligned to the center of the membrane in each experiment. Therefore, the horizontal position of the piezo transducer relative to the membrane was slightly different in each experiment. Variance in the frequency response could be caused by measuring slightly closer or further away from the nodes and anti-nodes generated by a resonant membrane. The effects of the vertical position of the piezo element relative to the center of the membrane and the vertical position of the optical fiber relative to the center of the membrane were not fully accounted for in this research. However, in future research, these parameters could be investigated by a finite element modeling method.

Another possible source of inconsistency between consecutive measurements could be related to the instability of the acoustic transducer. To test it, we measured the displacement versus voltage values of the piezoelectric transducer multiple times (>4 times), as shown in Figure 8. According to this graph, the stability and the repeatability of the transducer are quite good; therefore, the contribution of this factor to the measurement inconsistency is minimal.

Although completely consistent spectra were not obtained in this work, certain membranes did display a repeatable frequency peak. For example, the Silison A membrane showed a repeatable peak at 9 kHz, whereas the Norcada membranes showed a repeatable frequency peak at 46 kHz. Surprisingly, this result differed from the theoretical resonance frequency of 80 kHz for the Norcada membranes and the resonance of a Norcada membrane measured by Pearson et al. at 78 kHz [25]. Furthermore, the 46 kHz peak had a quality factor of 5.8, in contrast to the quality factor obtained by Pearson et al. of 1.2 ± 0.1 × 10^3^ [25]. The largest measured frequency response of the Norcada membrane was at 168 kHz with a Q of 33.6, which was below the aimed quality factor of 104. An explanation of the higher quality factor measured by Pearson et al. could be due to the use of a resonator tube. In these resonator tubes, a cavity is created, which keeps the sound waves confined to a small volume and enhances the strength of the sound waves through resonance. Resonator tubes were used for high-quality factor measurements of silicon nitride membranes in other research [25,26,27]. In our case, sound waves were generated in the open air. In a non-confined volume, such as air, sound waves dissipate proportionally to the square of the traveled distance. As a result of this dissipation, the sound waves hitting the membrane are much weaker than the sound waves in a resonator tube. The drop in quality factor described in our study might be attributable to the lack of a resonant tube. We attempted to use a resonant tube in our initial tests. However, the mechanism used to place the piezo element in the resonant volume caused a non-flat frequency response of the piezo. This response made it difficult to distinguish the resonant frequency of the membrane from the response of the piezo element. One can increase the quality factor of the membranes using different approaches. Since it is the ratio between the central frequency and the FWHM of the resonance peaks, it can be increased either by shifting the frequency of the resonance toward higher values by using a thinner membrane, a membrane with a higher Young’s modulus value, or by decreasing the FWHM of the resonance through a better coupling between the membrane and the acoustic field (reducing loss). 

We finally tried to detect methane with the ZonaSens interferometer using a Norcada membrane as the sensor. However, the results from this experiment indicated no significant difference from the background. The sound waves generated by the methane molecules were likely much weaker than the sound waves generated by the piezo element. As a result, the vibration of the Norcada membrane caused by sound waves from the excitation of the methane molecules was below the detection limit of the ZonaSens interferometer. Lowering the detection limit further might be difficult, as the ZonaSens interferometer was designed for interrogating deflections in optical fibers using FBGs instead of silicon nitride membranes. As a result, the software of the ZonaSens had difficulty with maintaining a linearized signal. Instead, enhancing the signal strength might be a more viable alternative. This could be achieved by using the previously mentioned resonator tube to enhance the power of sound waves through resonance. A different approach to increasing the signal strength would be to use a more intense laser source or to lower the loss in the optical fiber that guides the excitation laser. An increase in laser power reaching the methane molecules would result in a stronger sound wave reaching the membrane. However, the light source we use in the methane measurements has a fixed maximum power level; therefore, we could not test the effect of laser power and determine whether the system was feasible in the detection part. Alternatively, one can also choose a stronger methane resonance in the mid-infrared range; however, this will require a more expensive laser source and special fibers. 

## 5. Conclusions and Outlook

In conclusion, we investigated the potential of silicon nitride membranes in fiber-optic photoacoustic sensing. We used two different types of membranes produced by two different vendors and measured their frequency spectra using interferometry-based measurement systems. Differences in the spectra were observed for the same membranes measured at different times. Certain repeatable peaks in the frequency response were observed; however, the frequency of these peaks deviated from the expected resonance frequency and had a low-quality factor. Methane detection was attempted but did not result in a signal with a significant difference from the background. An alternative for further research is using a reflective cantilever as a microphone for the optical fiber-based photoacoustic detection of methane. Optical cantilevers have shown increased sensitivity compared to capacitive microphones [28]. Moreover, optical cantilevers have been used in photoacoustic gas detection at low frequencies [29]. This research has shown that an optical fiber-based interferometer is capable of interrogating the frequency response of reflective thin silicon nitride membranes. Thus, reflective optical cantilevers could be a viable target for further investigation.

## Figures and Tables

**Figure 1 sensors-23-01207-f001:**
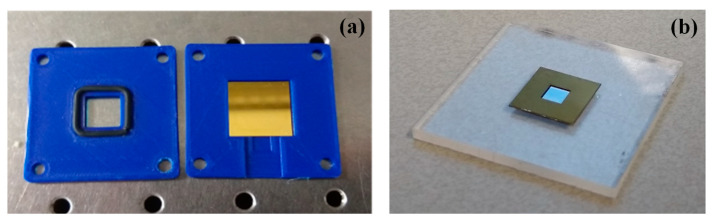
Silicon nitride membranes made by (**a**) Silison and (**b**) Norcada.

**Figure 2 sensors-23-01207-f002:**
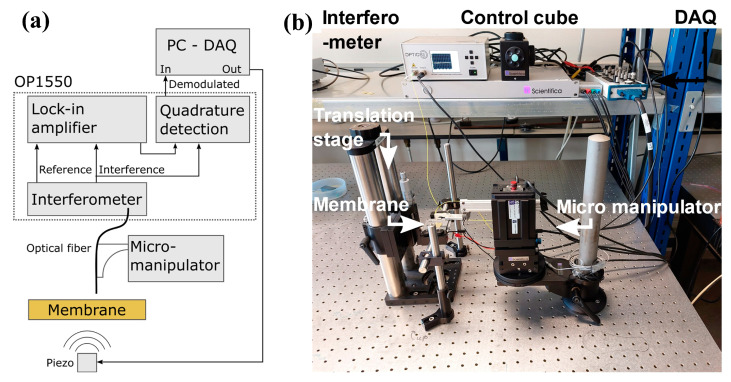
(**a**) Schematic of the setup and (**b**) a photograph of the setup. The piezo is positioned within a distance of approximately 8 mm from the membrane with the translation stage. A controller cube is used to control the micromanipulator and move the fiber close to the center of the membrane. The DAQ generates a sine wave that drives the piezo element for a specific duration and at a specific frequency and amplitude.

**Figure 5 sensors-23-01207-f005:**
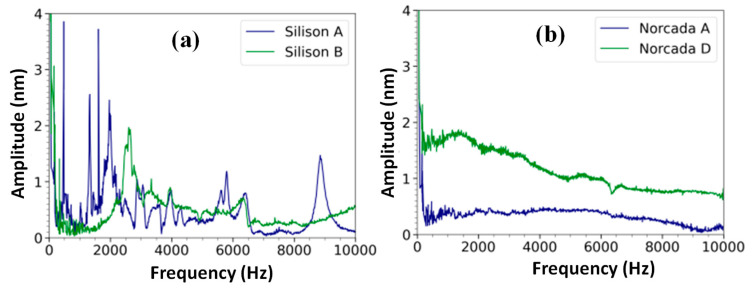
The frequency response of four different membranes measured at the center of the membrane is shown. (**a**) The spectra of Silison A and B membranes mounted on the washer plate. (**b**) The spectra of two Norcada membranes mounted on the o-ring. The piezo was used to generate a chirp pattern with a duration of 100 s at an amplitude of 10 Vpp. The distance from the piezo to the membrane was set at approximately 5 mm.

**Figure 6 sensors-23-01207-f006:**
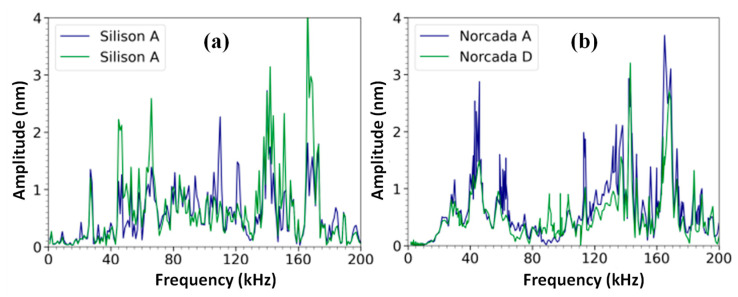
Acoustic spectra of the Silison A, Norcada A, and Norcada D membranes show multiple peaks over the frequency range of 0−200 kHz. (**a**) Repeated measurements of the Silison A membranes 2 h apart. (**b**) The Norcada A and Norcada D membranes show repeatable results on different days.

**Figure 7 sensors-23-01207-f007:**
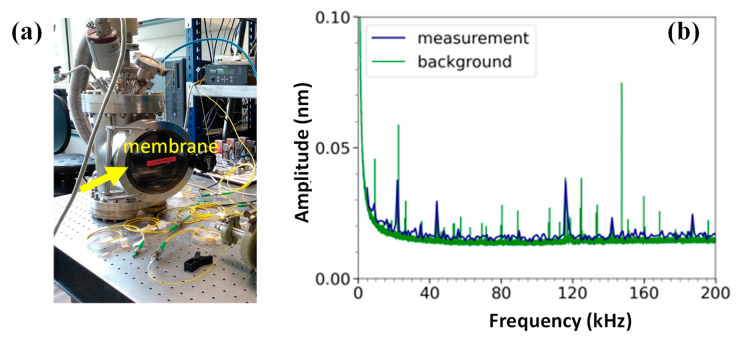
(**a**) The measurement setup used in methane detection experiments. (**b**) The frequency response of the membrane, which was inside the big chamber. The background is shown as a comparison.

**Figure 8 sensors-23-01207-f008:**
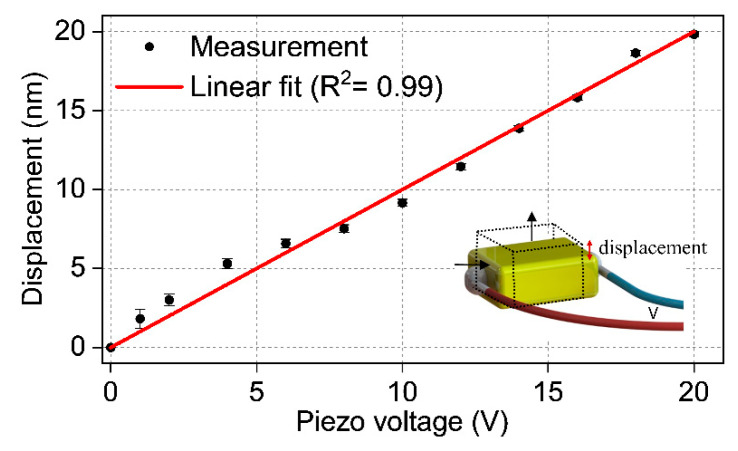
Measured displacement value of the piezo transducer at different voltage values. The inset shows the longitudinal displacement of the piezo when voltage is applied.

## Data Availability

The data presented in this study are available from the corresponding author upon reasonable request.

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
