# Peer review of "Investigating the Potential of Thin Silicon Nitride Membranes in Fiber-Based Photoacoustic Sensing"

_sensors, 2023, doi:10.3390/s23031207_

Round 1
Reviewer 1 Report
In “Investigating the Potential of Thin Silicon Nitride Membranes in Fiber-based Photoacoustic Sensing,” Konijn et al. describe calibration measurements on the responses of commercially-available silicon nitride membranes to acoustic excitation. These membranes are being investigated for purposes of determining their use as microphones in photoacoustic sensors for detection of methane in the atmosphere. The manuscript is well written and describes measurements that might be of interest to workers developing photoacoustic sensors. Even though the authors report a null result for detection of methane, this should not disqualify the work for further consideration. However, the manuscript generally fails to accurately represent the aims of the work, and the experiments that are reported are not designed to obtain results that might be used to make a significant report.
The overall aim of the work appears to be characterization of silicon nitride membranes for use as microphones in photoacoustic sensors, but the introduction focuses on larger-scale problems associated with photoacoustic measurements of methane levels in the atmosphere. In this respect, the authors fail to provide the necessary background for their work since photoacoustic sensing of trace gases in the atmosphere has been investigated extensively over the past 4+ decades (see work by Markus Sigrist and others) and it isn’t entirely clear that the microphone they are characterizing has advantages over previous designs.
With respect to the design of experiments, it seems the authors should have followed best practices for characterization and calibration of microphones. For example, observations regarding the effects of boundary conditions are interesting, but they beg additional documentation beyond the descriptions provided in the report. What were the distribution of stresses induced by the mounting configurations that were tested? Moreover, the authors didn’t report the stability of their acoustic source and it’s unclear how it might have affected various results they report.
Note that the axis labels in Figure 5 show frequencies up to 10 MHz, but this doesn’t appear to be consistent with the rest of the work. This report might be appropriate for a sensors-related conference, but the work and presentation in their current forms are not at the level that warrant publication in a science/engineering journal.
Author Response
Reviewer #1
- The overall aim of the work appears to be characterization of silicon nitride membranes for use as microphones in photoacoustic sensors, but the introduction focuses on larger-scale problems associated with photoacoustic measurements of methane levels in the atmosphere. In this respect, the authors fail to provide the necessary background for their work since photoacoustic sensing of trace gases in the atmosphere has been investigated extensively over the past 4+ decades (see work by Markus Sigrist and others) and it isn’t entirely clear that the microphone they are characterizing has advantages over previous designs.
We thank the reviewer for this remark. Actually, we started working on silicon membranes as an alternative approach to improve the detection sensitivity of the photoacoustic sensing concept called PAS-WRAP. This concept relies on wrapped optical fibers around a chamber, working like a guitar upon acoustic excitation (see Ref. 12: Zhou et al.). Even though this is a very promising and novel idea, the sensitivity is not sufficient for methane detection. In our current work, we considered several different microphone ideas that can be applied to the PAS-WRAP project, and among them, the thin membrane concept seemed to be the most promising according to the literature. However, we faced several practical issues, which were not properly reported in the literature. Therefore, in this manuscript, we wanted to share our experience and experimental outcomes with others as a reference for their future research.
In this respect, we changed our introduction by focusing more on the fiber-based PA sensors with additional references to background literature in the revised manuscript.
- With respect to the design of experiments, it seems the authors should have followed best practices for characterization and calibration of microphones. For example, observations regarding the effects of boundary conditions are interesting, but they beg additional documentation beyond the descriptions provided in the report. What were the distribution of stresses induced by the mounting configurations that were tested? Moreover, the authors didn’t report the stability of their acoustic source and it’s unclear how it might have affected various results they report.
We thank the reviewer for these questions. Although we could not measure the stress distribution for different mounting configurations, we attempted to simulate it using finite element modeling; however, the relatively big size of the membranes and their frames made it quite difficult to simulate the actual devices. Moreover, the exact mechanical properties of some of the components (frame, bolts, rubber spacing, etc.) are not known, which affects the accuracy of the results significantly. As common sense, when a single bolt is tightened, we expect that the maximum of the membrane stress shift towards the edge of the membrane where the corresponding bolt is closer. When all bolts are tightened in the same way, then the membrane stress is uniformly distributed over the membrane. In our experiments, we tried to create a uniform stress distribution over the membrane as repeatable as possible. However, we concluded that it is a quite challenging task. We added this information in the “Discussion” section.
We used a small piezoelectric transducer and measured its displacement versus voltage values multiple times on different days. A new plot is included in the manuscript, showing displacement versus voltage values of the piezo transducer. According to this graph, the stability and the repeatability of the transducer are quite good; therefore, it does not influence the experimental results. This information is included in the “Discussion” section.
- Note that the axis labels in Figure 5 show frequencies up to 10 MHz, but this doesn’t appear to be consistent with the rest of the work. This report might be appropriate for a sensors-related conference, but the work and presentation in their current forms are not at the level that warrant publication in a science/engineering journal.
There is a typo in the x-axis of Fig. 5. Instead of kHz, it should have been Hz. We changed it in the revised manuscript. As the reviewer mentioned earlier, this work can be a good reference for those who are planning to work on such experimental work. As a personal side note, Dr. Akca worked on acoustic excitation of biological corneas during her postdoctoral research and she faced similar experimental difficulties related to boundary condition settings of the corneas. Still, there is no easy way of solving this problem and we believe our work can at least provide some insight into this common issue. In this respect, publishing our work (after implementing the requested improvements) as a journal paper rather than a conference paper will increase its visibility and impact.
Reviewer 2 Report
Review of the manuscript titled " Investigating the Potential of Thin Silicon Nitride Membranes in Fiber-based Photoacoustic Sensing " by Konijn et al.
The work done by the author is interesting and easy to understand the work. The author gives satisfactory reasons for not using a silicon nitride membrane as a sensor. The work is a little bit confusing as on the one hand this work is done by previous authors who use silicon nitrate as a sensor and now the author told some restrictions about it. From there I have some major comments and suggestions to improve the work.
1- What is a full form of PAS-WRAP? The author will give more details about the PAS trace gas sensor and in the introduction section, some more references could be added.
2- Where is testing about the detection limit of the membranes in a methane detection experiment?
3- What is the difference between Silison A, Silison B, Norcada A, and Nocrada D membranes?
4- Can we calculate the resonance frequency of the second, third, etc. modes using PAS?
5- In the measurement setup author is not giving details about the lock-in amplifier. How is the author changed the frequencies of the measurements? The author could also give more details about the experimental setup.
6- How could the author decide the peaks having multiple data in a spectrum? How could the author find out the signal-to-background ratio?
7- How could the author explain the spectral peaks presented in different membranes?
8- Which process you can increase the quality factor of samples?

Author Response
Reviewer #2
- What is a full form of PAS-WRAP? The author will give more details about the PAS trace gas sensor and in the introduction section, some more references could be added.
We thank the reviewer for this suggestion. We changed the Introduction by including more information on the PAS-WRAP concept and one more reference.
- Where is testing about the detection limit of the membranes in a methane detection experiment?
We thank the reviewer for this question. As we discussed in the sub section “3.5. Methane detection”, we did not get a strong signal (not different than the background noise); therefore, we could not continue with the detection limit experiments.
- What is the difference between Silison A, Silison B, Norcada A, and Nocrada D membranes?
Silison A and Silison B are the membranes, which were produced by Silison and they have similar mechanical and physical properties. Norcada A and Norcada D were produced by Norcada and they also have similar properties. The rest of their properties were discussed in the subsection “Membranes and membrane holders”. We updated this section by including more information in the revised manuscript.
- Can we calculate the resonance frequency of the second, third, etc. modes using PAS?
We thank the reviewer for this question; however, it is not very clear to us what the reviewer meant by this question. PAS is not a technique to calculate resonance frequencies. We measure the resonance frequencies of a membrane and by using an excitation laser that is modulated at one of these resonance frequencies, we create PA effect and use it for gas detection. If the reviewer meant whether higher-order resonances can be used for PAS detection, the answer is yes. It will only change the modulation frequency of the excitation laser.
- In the measurement setup author is not giving details about the lock-in amplifier. How is the author changed the frequencies of the measurements? The author could also give more details about the experimental setup.
The lock-in amplifier is a built-in component of the OP1550 system, which was developed by Optics11. By scaling the lock-in signal and interferometric readout a linear relation is obtained to the membrane deflection. The phase can be obtained by computing the angle between the scaled interferometric readout voltage and scaled lock-in voltage. The DAQ generates a sine wave that drives the piezo element for a specific duration and at a specific frequency and amplitude. We included this information to the “Measurement Setup” section.
- How could the author decide the peaks having multiple data in a spectrum? How could the author find out the signal-to-background ratio?
We took each spectrum with minimum possible frequency step (~10 Hz) to be able to resolve peaks with higher details.
The SNR is calculated by dividing the amplitude of the signal at the resonance frequency by the amplitude of the background that is relatively far from the resonance peak.
These information was included in the revised manuscript.
- How could the author explain the spectral peaks presented in different membranes?
The spectral speaks represent the eigenfrequencies of the membrane. Some of them are weaker than others but they can still be excited. This information is included in the “Discussion” part of the revised manuscript.
- Which process you can increase the quality factor of samples?
The quality factor (Q) is the ratio between the central frequency and the FWHM of the resonance peaks. The Q value can be increased either by shifting the frequency of the resonance towards higher values through a thinner membrane or a membrane with a higher Young’s modulus value or decreasing the FWHM of the resonance through a better coupling between the membrane and the acoustic field (reducing loss). This information is included in the “Discussion” section.
Reviewer 3 Report
To the author:
This manuscript demonstrates the potential of different kinds of silicon nitride membranes used for photoacoustic spectroscopy. Starting from the resonance spectrum measurement, the author compared the resonance peaks under different cases and achieved some important conclusions. For example, under O-ring mount, the pressure exerted on the frame greatly affects the peak positions. While under washer ring mount, 9-kHz peak repeatedly appear in the results at different times. In addition, OP1550 interferometer and ZonaSens interferometer were used to measure the resonance spectrum respectively. As a research article, this article can provide readers with some ideas about new sensing methods. However, from the point of view of experimental results, the index breakthrough cannot be achieved. Repeatability and practicality are the weaknesses. In this sense, I suggest that reconsider after revision. The following is the revision suggestions and comments:
1. Could the resonance spectrum be measured in a closed gas chamber to avoid the influence of room temperature variation?
2. In the methane detection part, could the author adopt higher power excitation light to realize methane sensing? In this way, we can determine whether the system is feasible in the detection part.
3. What do the three curves in Figure 3b represent? Maybe it shows the resonance spectrum at different intensities of sound waves, but it’s better to indicate it in the text or the legend.
4. On line 159, it’s better to add “as shown in Figure 4c”. Similarly, please indicate what the three curves in Figure 4c represent respectively in the text.
5. Does "0.2% CH4" on line 56 refer to Ref. 12? Actually, Ref. 12 does not refer to the measurement of methane, but acetylene with the detection limit of 24 ppb. Hope the author can explain.
6. The x-axis in Figure 5 should be Hz, rather than kHz.
Author Response
Reviewer #3
- Could the resonance spectrum be measured in a closed gas chamber to avoid the influence of room temperature variation?
We thank the reviewer for this interesting suggestion. We have a custom-made closed gas chamber, which is too small to fit in all parts of the measurement setup. It was originally designed to place only the fiber-based PA sensor part, which is a small part of the overall setup. Even though we can’t perform these measurements, it is still a valuable suggestion to include in the discussion part.
- In the methane detection part, could the author adopt higher power excitation light to realize methane sensing? In this way, we can determine whether the system is feasible in the detection part.
We thank the reviewer for this useful comment. We have also considered this possibility; however, the laser we used for the photoacoustic excitation has a fixed max power level and a more powerful light source is not available in our lab at the moment. Alternatively, stronger vibrational modes in the mid-IR range can be excited, which requires an expensive light source and compatible fibers that are much more lossy and expensive as well. We included this information in the “Discussion” section.
- What do the three curves in Figure 3b represent? Maybe it shows the resonance spectrum at different intensities of sound waves, but it’s better to indicate it in the text or the legend.
The three curves in Fig. 3b indicate three consecutive membrane spectrum measurements under the same boundary settings. This point is clarified in the revised manuscript as well as in the figure legend.
- On line 159, it’s better to add “as shown in Figure 4c”. Similarly, please indicate what the three curves in Figure 4c represent respectively in the text.
We added the suggestion of the reviewer on line 160. The three curves in Fig. 4c indicate three consecutive membrane spectrum measurements. This point is clarified in the revised manuscript.
- Does "0.2% CH4" on line 56 refer to Ref. 12? Actually, Ref. 12 does not refer to the measurement of methane, but acetylene with the detection limit of 24 ppb. Hope the author can explain.
We thank the reviewer for this remark. The information was correct but the reference was wrong. The correct one is: Zhou S., Slaman M., Gruca G. and Iannuzzi D. “PAS-WRAP: a new approach to photoacoustic sensing, a new opportunity for the optical fiber sensor community”. In: Seventh European Work- shop on Optical Fibre Sensors. Vol. 11199. International Society for Optics and Photonics. 2019, 111992S. We changed Ref. 12 with the correct reference in the revised manuscript.
- The x-axis in Figure 5 should be Hz, rather than kHz.
We thank the reviewer for pointing this typo, which we corrected in the revised manuscript.
Round 2
Reviewer 1 Report
The work focuses on characterizing a type of microphone for photoacoustic (PA) sensing of gases in the atmosphere, but the authors have not provided adequate background on PA sensing for gas detection. They need to review and cite work by M. Sigrist, and compare the performance of their microphones to those used in previous work.
Author Response
We thank the reviewer for this comment. In the earlier revised version, we already cited one relevant work from Prof. Sigrist (Ref. 11). In the current revised version, we included three more publications from Prof. Sigrist’s group related to PA sensing and gas detection (as listed below), and also discussed the new concepts introduced by his group together with the microphone concepts used in each. This new information was included in the “Introduction” section.
Bartlome R., Kaučikas M. and Sigrist M. W. Modulated resonant versus pulsed resonant photoacoustics in trace gas detection, Applied Physics B, 2009, 96, 561-566.
Sigrist M. Photoacoustic spectroscopy in trace gas sensing, 2nd International Conference on Physics of Optical Materials and Devices (ICOM 2009) August 27-30, 2009.
Rey J.M. and Sigrist M.W. Differential mode excitation photoacoustic spectroscopy: A new photoacoustic detection scheme, Review of Scientific Instruments, 2007, 78, 063104.
Reviewer 2 Report
The author incorporated all the suggestions. This study can provide new ideas in the detection of PAS-WRAP. I urge the editor to accept the manuscript after some minor revisions.
Author Response
We thank the reviewer for his/her valuable comments that helped us to improve our manuscript further. We corrected some of the typos and grammar mistakes in the current version as suggested by the reviewer. We have extended the background given in the Introduction section by including 3 more works by Prof. M. Sigrist.
Reviewer 3 Report
Although gas absorption could not be measured successfully, but the study of the properties of membrane, can provide some new ideas for photoacoustic spectroscopy. The paper is worthy of publication, but need some minor revisions. The introduction focuses on the study of photoacoustic spectroscopy based on methane, which seems to be inconsistent with the main idea of this paper. It is suggested to start with the sensing method of photoacoustic spectrum for technical discussion and parameter comparison.
Author Response
We thank the reviewer for his/her suggestion. We have been working on the PAS-WRAP concept for more than 3 years, and our motivation was always to detect methane because of its harm to the environment. This is why we started with the big picture and narrowed it down to our concept throughout the introduction. Even though our manuscript reports our findings on microphone design, the ultimate goal is still the same. We respect the suggestion of the reviewer, but we believe that the current introduction is more appropriate to emphasize the importance of our study. We hope the reviewer can accept our reasoning.